# CAN LLMS SERVE AS CAUSAL INFERENCE AGENTS? A STUDY ON POST-TRAINING METHODS

## ABSTRACT

Causal inference is essential for decision-making but remains challenging for non-experts. While large language models (LLMs) show promise in this domain, their precise causal estimation capabilities are still limited, and the impact of post-training on these abilities is insufficiently explored. This paper examines the extent to which post-training can enhance LLMs' capacity for causal inference. We introduce `CausaGym`, a comprehensive dataset comprising seven core causal tasks for training and five diverse test sets. Using this dataset, five post-training approaches—SFT, DPO, KTO, PPO, and GRPO are systematically evaluated. Across five in-domain and four existing benchmarks, our experiments demonstrate that appropriate post-training enables smaller LLMs to perform causal inference competitively, often surpassing much larger models. Our 14B-parameter model achieves 93.5% accuracy on the CaLM benchmark, compared to 55.4% by OpenAI o3. Furthermore, the post-trained agents exhibit strong generalization and robustness under real-world conditions such as distribution shifts and noisy data. Collectively, these findings provide the first systematic evidence that targeted post-training can produce reliable and robust LLM-based causal inference agents.

## 1 INTRODUCTION

Causal inference, a core component of human cognition, seeks to distinguish causation from association by estimating their effects between variables (Pearl, 2009; Sloman & Sloman, 2009). Causal inference is crucial because decision-makers must both predict intervention effects and evaluate counterfactual outcomes (Woodward, 2005; Shpitser & Pearl, 2006; Bunge, 2017; Chen et al., 2024c). For example, one may estimate how deploying a treatment changes population health and what the same patients' outcomes would have been had they not been treated (Pearl & Mackenzie, 2018).

Many statistical methods have been developed for causal inference with observational data (Pearl, 2010). Broadly, these methods recover causal effects either by approximating randomized assignment via adjustment for measured confounders or by exploiting quasi-experimental variation that makes treatment as-if random (Rubin, 1974; 2005; Pearl et al., 2016). To facilitate the application of these methods, there has been a surge in the development of new causal inference libraries (Battocchi et al., 2019; Sharma & Kiciman, 2020; Chen et al., 2020). They encapsulate complex algorithms, providing researchers with systematic tools for analysis and lowering the barrier to applying causal inference. Despite lowering entry barriers, these libraries remain difficult to use correctly for non-experts. One must still articulate an identification strategy, verify assumptions, and interpret diagnostics. This challenge naturally leads to the question of whether we can develop a *causal inference agent* that explains its assumptions and reasoning in plain language, making the causal inference process fully auditable.

LLMs appear promising for addressing this challenge. Their natural-language interfaces can help non-experts articulate identification questions, surface assumptions, and obtain step-by-step explanations of anal-

yses. And they have shown striking performance on tasks requiring complex reasoning, e.g., mathematics (Luo et al., 2025), coding (Nam et al., 2024), and formal theorem proving (Quan et al., 2024). Despite these strengths, studies show that LLMs still struggle with causal inference—especially when precise numerical estimation is required (Jin et al., 2023; Chen et al., 2024b; Jin et al., 2024). Moreover, some work suggests that LLMs may be inherently incapable of performing formal causal reasoning (Chi et al., 2024). Although various post-training methods have proven effective in enhancing the reasoning and agent capabilities of LLMs (Wang et al., 2024; Chen et al., 2024d; Song et al., 2024; Guan et al., 2025; Guo et al., 2025), no systematic research has yet explored whether—and to what extent—these gains transfer to causal inference. Therefore, this paper addresses this gap by asking:

> *Can LLMs become effective causal inference agents through post-training?*

To address this question, a training corpus was constructed to cover seven causal inference tasks (Rubin, 2005; Pearl et al., 2016): average treatment effect (ATE), controlled direct effect (CDE), effect of the treatment on the treated (ETT), natural direct effect (NDE), natural indirect effect (NIE), probability of necessity (PN), and probability of sufficiency (PS). Together, these tasks span both interventions and counterfactuals, enabling a comprehensive strengthening of an agent's causal inference capabilities (Chen et al., 2024b;a).

Then, three categories of post-training methods are evaluated: supervised fine-tuning (SFT), offline reinforcement learning (RL), and online RL. These categories encompass five representative algorithms: SFT includes vanilla SFT (Wei et al., 2022); offline RL includes Direct Preference Optimization (DPO) (Rafailov et al., 2023) and Kahneman–Tversky Optimization (KTO) (Ethayarajh et al., 2024); online RL includes Proximal Policy Optimization (PPO) (Schulman et al., 2017; Ouyang et al., 2022) and Group Relative Policy Optimization (GRPO) (Shao et al., 2024).

Finally, the agents are evaluated across nine test sets to assess their causal inference capabilities, generalization (i.e., performance under distribution shift), internalization (i.e., whether the agent truly understands the underlying causal inference theorems), and robustness to practical stressors (i.e., noise and missing data) relevant to real-world settings.

The comprehensive experiments on nine diverse testing sets using DeepSeek-R1-Distill-Qwen-14B demonstrate that proper post-training can enable smaller-scale LLMs to function as strong *causal inference agents* that surpass the larger-scale LLM. Specifically, GRPO emerges as the most effective method, achieving an impressive 93.5% accuracy on the CaLM benchmark (Chen et al., 2024b), whereas DeepSeek-R1-0528-671B and OpenAI o3 reach only 57.0% and 55.4%, respectively. Moreover, the post-trained agents not only excel at causal inference but also exhibit strong generalization to distribution shift, effective internalization of knowledge, and robustness to noise.

To summarize, the main contributions of this paper are:

1. To the best of our knowledge, this is the first work to thoroughly investigate the effects of current mainstream post-training methods on the causal inference abilities of LLMs.

2. We introduce the `CausaGym` dataset, comprising (i) the first training set designed to systematically enhance LLMs as *causal inference agents*. This training set encompasses seven distinct tasks and is adaptable to five different post-training methods. And (ii) a suite of five test sets that evaluate agents along three dimensions: generalization, internalization, and robustness.

3. We conduct comprehensive experiments to validate the causal inference capabilities, generalization, internalization, and robustness of agents built using various post-training methods.

## 2 METHODOLOGY

Our method proceeds in four main steps. First, we generate training data from synthetic SCMs (Sec. 2.1) and create five specialized testing sets to thoroughly evaluate the *causal inference agent*'s ability (Sec. 2.2). We collectively refer to the entire corpus as `CausaGym`. Next, we adopt a two-stage training strategy: cold-starting the LLM with SFT on a small amount of data (Sec. 2.3), followed by five various post-training methods (i.e., SFT, PPO, GRPO, DPO and KTO) to enhance the LLM's causal inference ability (Sec. 2.4).

### 2.1 TRAINING DATASET GENERATION

Our approach is two-fold: we first generate a base dataset and then make fine-grained adjustments according to the requirements of each post-training method.

#### 2.1.1 BASE DATASET CONSTRUCTION

While the focus of Chen et al. (2024b) is the testing set, the field still lacks sufficient training data for causal inference. We address this by replicating their data construction method and using it as a foundation to build extended training sets tailored for various post-training methods. To achieve this, we employ the following four steps to construct our base dataset:

**Step 1: generating DAGs.** We create the backbone structure for our SCMs by randomly generating 10-node DAGs, a graph size widely adopted in current research (Jin et al., 2023; Chen et al., 2024b;c).

**Step 2: semantifying nodes.** Following Jin et al. (2023) and Chen et al. (2024b), we assign meaning to the nodes in three distinct ways to ensure diversity: (a) **Real:** Nodes receive semantically meaningful labels, and their causal relationships are coherent based on domain knowledge. (b) **Random:** Nodes receive semantically meaningful labels, but the causal relationships between them are randomized. (c) **Fake:** The nodes' labels are assigned as stochastic four-letter strings.

**Step 3: determining SCMs.** We model the underlying functions of the SCMs with single-layer perceptrons, aligning the defined causal relationships with the perceptron parameters. To be specific, the SCM function for a node $X$ can be written as:

$$X := f_X \left( PA_X^1, PA_X^2, \ldots, PA_X^k, U_X \right) \tag{1}$$

$$= \begin{cases} 0, & U_X - g_X \left( PA_X^1, PA_X^2, \ldots, PA_X^k \right) > 0 \\ 1, & \text{otherwise} \end{cases} \tag{2}$$

where $PA_X^i$ denotes the parent nodes of X, $U_X$ is an independent random variable uniformly distributed from $[0,1]$ and $g_X : \{0,1\}^k \to [0,1]$ is a function to be determined. In this paper, we model it by a single-layer perceptron as follows:

$$g_X \left( PA_X^1, PA_X^2, \ldots, PA_X^k \right) = \text{sigmoid}(b_X + \sum_{i=1}^{k} w_X^i PA_X^k). \tag{3}$$

Both $b_X$ and $w_X^i$ are generated randomly, and we ensure the sign of $w_X^i$ is consistent with the assigned relationship between X and $PA_X^i$.

**Step 4: generating instances.** We approximate the required probabilities (marginal and conditional) by sampling from the SCMs. Utilizing this data and predefined templates, we generate questions, answers, and symbolic solutions for seven distinct causal tasks.

Figure 1 provides an example of the resulting data. We then leverage `DoWhy` (Sharma & Kiciman, 2020) to identify the necessary backdoor adjustment set and mediator set, and then combine this information to formulate the symbolic solutions with several templates[1]. At this point, the base dataset is complete and can be processed into the specific formats needed for all subsequent post-training methods.

**Base Dataset**

Problem:

Input Info: Imagine a self-contained, hypothetical world with only the following conditions, and without any unmentioned factors or causal relationships: Traffic congestion has a direct effect on appearance. Traffic congestion has a direct effect on rainfull. Appearance has a direct effect on temperature.
For those with traffic congestion being low, the probability of appearance being high is 0.4358. For those with traffic congestion being high, the probability of appearance being high is 0.0511.

Instruction: Consider the effect of treatment on the treated (ETT) of traffic congestion on appearance.

Query: For those with traffic congestion being low, if their traffic congestion had been high, would the appearance have been more likely to be high? Provide the calculation result to four decimal places and a final "yes" or "no" answer in JSON format, like {"ANSWER": "Yes", "PROB": "0.1234"}:

Step-by-step symbolic solution:
Step 1: Check whether treatment (A) is a cause of outcome (B). Node A is a cause of node B because there is one or more directed paths from A to B (e.g. A->B).
Step 2: Identification of the Causal Effect. P(B|do(A)) can be identified.
Step 3: Find a valid backdoor adjustment set: empty set.
Step 4: ETT = E[B_{A=0} - B_{A=1} | A=0] = P(B=1|A=0) - P(B=1|A=1)
Step 5: P(B=1|A=0) = 0.4358, P(B=1|A=1) = 0.0511
Step 6: ETT = 0.4358 - 0.0511 = 0.3847
Step 7: ETT = 0.3847 > 0 so the answer is No.

Ground Truth: { "ANSWER": "Yes", "PROB": "0.3847"}

Figure 1: An example of the base dataset.

### 2.1.2 METHOD-SPECIFIC ADAPTATION

We now proceed to describe how the base dataset is modified for various post-training methods.

(1) **SFT.** For each question, we first provide a step-by-step symbolic solution to prompt the DeepSeek-R1-0528-671B to generate a correct and naturally phrased reasoning process and answer. The SFT dataset retains these rephrased reasoning traces and correct answers for finetuning. (2) **Offline RL.** On the same set of questions, we construct paired positive/negative reasoning samples. (a) Positive: Built in the same way as SFT. (b) Negative: Generated by instructing the DeepSeek-R1-0528-671B to answer questions without any step-by-step guidance. If the final answer is wrong, we use the associated reasoning as the negative sample. If the model consistently answers correctly, we select an unguided reasoning trace that is verbose, missing key steps, or misaligned with the standard solution as the negative sample. Accordingly, DPO forms one positive–negative pair per question, while KTO collects all generated positive and negative samples and enforces a 1:1 ratio. (3) **Online RL.** Since GRPO and PPO rely solely on the final-answer reward and format reward (reasoning format and json format), chain-of-thought annotations are unnecessary. For these online RL methods, we only need to extract the questions and final answers directly from the base dataset.

### 2.2 TESTING DATASET GENERATION

To thoroughly evaluate a *causal inference agent*'s ability, we create five specialized testing sets based on three distinct perspectives: (1) Generalization. To test whether the agent truly understands the meaning of the questions rather than simply memorizing the specific phrasing found in the training data, we design the `CausaGym`-rephrased. (2) Internalization. To determine if the agent truly understands the underlying causal inference theorems rather than relying on superficial shortcuts, we create the `CausaGym`-omitted and `CausaGym`-deconfounding. (3) Robustness. To assess the agent's robustness under the non-ideal, noisy data conditions common in the real world, we construct the `CausaGym`-redundantand `CausaGym`-insufficient. To build these new testing sets, we take the ATE, CDE, ETT, NDE, NIE, PN, and PS tasks from the original CaLM dataset (Chen et al., 2024b) and modify them in various ways.

**CausaGym-rephrased.** The `CausaGym`-rephrased is constructed by prompting the DeepSeek-R1-0528-671B to rephrase the whole problem. The meaning, probabilities, and step-by-step solving procedure remain preserved. An example is shown in Figure 6.

---

[1]To be specific, the predefined templates for questions are shown in Table 3

**CausaGym-omitted and CausaGym-deconfounding.** The `CausaGym`-omitted paraphrases the original whole problem and preserves its semantics and probabilities, but intentionally omits the instruction part (previously shown in Figure 1) that would otherwise reveal the specific task type. This design prevents models from relying on explicit task cues. The questions in the `CausaGym`-deconfounding can only be solved by applying the backdoor criterion. We specifically avoid cases such as those where no causal relationship exists between the cause and effect, or where confounders are absent. This design ensures that the model cannot rely on spurious correlations to get the correct answer and must understand the underlying causal structure. The examples are shown in Figure 7 and 8.

**CausaGym-redundant and CausaGym-insufficient.** In the real world, causal inference problems posed by non-experts are highly likely to contain either redundant or insufficient information. To test robustness against this, we construct `CausaGym`-redundant(by adding two correct but useless conditions) and `CausaGym`-insufficient(by removing two necessary conditions). We hypothesize that a true *causal inference agent* will successfully disregard the redundant data in `CausaGym`-redundant and identify the key missing information in `CausaGym`-insufficient. The examples are in shown in Figure 9 and 10.

## 2.3 COLD START

Before further employing other post-training methods, we first conduct SFT to cold start the base model. The optimization objective is shown as follows:

$$\mathcal{J}_{\text{SFT}}(\theta) = \mathbb{E}_{(x,y)\sim D}[\log \pi_\theta(y|x)], \tag{4}$$

where $\pi_\theta$ is the current model policy, $D$ is the dataset, $x$ is the input, and $y$ is the target completion.

## 2.4 POST-TRAINING METHODS

Online RL methods update the model by using feedback on its own rollouts. Both PPO and GRPO learn a policy by estimating advantages for state–action pairs and ascending the surrogate objective to maximize expected rewards. Mathematically, they aim to maximize the following function:

$$\mathcal{J}(\theta) = \mathbb{E}_{o_t, a_t \sim \pi_{\theta_{\text{old}}}} \left[ \frac{1}{G} \sum_{i=1}^{G} \frac{1}{|o_i|} \sum_{t=1}^{|o_i|} \min\left( r_t(\theta)\hat{A}_{i,t}, \text{clip}(r_t(\theta), 1-\epsilon_{low}, 1+\epsilon_{high})\hat{A}_{i,t} \right) \right]$$
$$- \beta D_{KL}(\pi_\theta|\pi_{\text{ref}}), \tag{5}$$

where $\pi_{ref}$, $\pi_\theta$ and $\pi_{\theta_{old}}$ are reference, current and old policy, $A_{i,t}$ is an estimated advantage, $r_t(\theta)$ is the log-likelihood ratio and $\epsilon_{low}, \epsilon_{high}$ are clip ratio high and low. The key distinction between PPO and GRPO lies in how the advantage is calculated: PPO uses a separate critic network to estimate the advantage, while GRPO uses the mean reward of the responses generated by the current policy.

DPO and KTO use a fixed, pre-collected dataset and train the LLM to separate good from bad behavior. Their optimization objectives are shown as follows:

$$\mathcal{J}_{\text{KTO}}(\theta) = \mathbb{E}_{(x,y)\sim D_{\text{desirable}}} \left[ \lambda_d \left( 1 - \text{sigmoid}\left( \beta \log \frac{\pi_\theta(y|x)}{\pi_{\text{ref}}(y|x)} - z_{\text{ref}} \right) \right) \right]$$
$$+ \mathbb{E}_{(x,y)\sim D_{\text{undesirable}}} \left[ \lambda_u \left( 1 - \text{sigmoid}\left( z_{\text{ref}} - \beta \log \frac{\pi_\theta(y|x)}{\pi_{\text{ref}}(y|x)} \right) \right) \right], \tag{6}$$

$$\mathcal{J}_{\text{DPO}}(\theta) = \mathbb{E}_{(x,y_w,y_l)\sim \mathcal{D}} \left[ \log \sigma\left( \beta \log \frac{\pi_\theta(y_w|x)}{\pi_{\text{ref}}(y_w|x)} - \beta \log \frac{\pi_\theta(y_l|x)}{\pi_{\text{ref}}(y_l|x)} \right) \right], \tag{7}$$

where $z_{\text{ref}}$ is the reference point, $\lambda_d, \lambda_u, \beta$ are hyperparameters.

## 3 EXPERIMENT

### 3.1 SETUP

**Baselines.** We consider a wide range of baselines, including Llama-3.3-70B (Meta, 2024), Qwen3-235B (Yang et al., 2025), DeepSeek-R1-Distill-Qwen-14B (Guo et al., 2025), DeepSeek-R1-0528-671B (Guo et al., 2025), Gemini 2.5 Pro (Deepmind, 2025), OpenAI o3 (OpenAI, 2025).

**Datasets.** Our evaluation covers a total of nine datasets: the five novel datasets we constructed in Sec. 2.2, the lite version of CaLM (Chen et al., 2024b) (focusing on numerical tasks for ATE, CDE, ETT, NDE, NIE, PN, and PS), and three external math benchmarks: Math 500 (Hendrycks et al., 2021), Minerva Math (Lewkowycz et al., 2022), and AMC 2023 (AMC, 2023).

**Prompts.** We employ a basic COT prompt (i.e., `<question, Let's think Step by Step>`) for all test sets. For `CausaGym`-insufficient, we further augment the prompt to explicitly instruct the LLM: *If the condition is not enough to solve the question, output 'LACK_CONDITION' as final answer.*

**Metrics.** The evaluation metric is accuracy. All questions are assessed with exact-match scoring.

**Implement details.** For SFT, we train 3 epochs on 3500 samples with lora (Hu et al., 2022). For DPO, we train 3 epochs on 3500 preferred and dis-preferred pairs. The hyperparameter $\beta$ is set to 0.1. For KTO, we train 3 epochs on 7000 samples with preference labels. The hyperparameter $\beta$ is set to 0.1, $\lambda_d$ and $\lambda_u$ are both set to 1. For GRPO, we train for three epochs on a dataset of 3500 questions. We set $\beta$ to 0, $\epsilon_{low}$ to 0.2, and $\epsilon_{high}$ to 0.28 and use rejection sampling (Yu et al., 2025). For PPO, we train for three epochs on a dataset of 3500 questions. We set $\beta$ to 0.001, $\epsilon_{low}$ to 0.2, and $\epsilon_{high}$ to 0.28 and use rejection sampling.

### 3.2 MAIN RESULTS

| LLM | ATE | CDE | ETT | NDE | NIE | PN | PS | Avg. |
|---|---|---|---|---|---|---|---|---|
| Llama-3.3-70B | 0.572 | 0.372 | 0.288 | 0.430 | 0.200 | 0.010 | 0.010 | 0.269 |
| Qwen3-235B | 0.004 | 0.000 | 0.180 | 0.230 | 0.000 | 0.000 | 0.000 | 0.059 |
| DeepSeek-R1-0528-671B | 0.740 | 0.540 | 0.220 | 0.460 | 0.450 | 0.780 | 0.800 | 0.570 |
| Gemini 2.5 Pro | 0.760 | 0.710 | 0.320 | 0.590 | 0.470 | 0.240 | 0.050 | 0.448 |
| OpenAI o3 | 0.840 | 0.590 | 0.300 | 0.430 | 0.720 | 0.450 | 0.550 | 0.554 |
| DeepSeek-R1-Distill-Qwen-14B | 0.594 | 0.364 | 0.210 | 0.442 | 0.212 | 0.014 | 0.066 | 0.272 |
| Cold Start Base | 0.634 | 0.550 | 0.156 | 0.294 | 0.434 | 0.788 | 0.714 | 0.510 |
| SFT | 0.852 | 0.828 | 0.470 | 0.560 | 0.604 | 0.858 | 0.766 | 0.702 |
| DPO | 0.656 | 0.514 | 0.198 | 0.282 | 0.510 | 0.806 | 0.708 | 0.524 |
| KTO | 0.716 | 0.674 | 0.232 | 0.412 | 0.472 | 0.812 | 0.700 | 0.574 |
| PPO | 0.972 | 0.982 | 0.806 | 0.926 | 0.924 | **0.940** | **0.902** | 0.921 |
| GRPO | **0.990** | **0.994** | **0.900** | **0.940** | **0.930** | 0.928 | 0.866 | **0.935** |

Table 1: Comparison of different post-training methods and a wide range of baselines. Best results are in **bold**, the second best results are underlined.

Table 1 presents a comparison of our different training approaches with baseline models. We draw the following conclusions: (1) Through appropriate post-training, it is possible to build *causal inference agents* using smaller-scale LLMs that outperform larger-scale LLMs. We exclusively train the *causal inference agent* on a 14B-scale LLM (i.e., DeepSeek-R1-Distill-Qwen-14B). Among these methods, DPO performs the least effectively, achieving a score of only 52.4%, while the best-performing GRPO reaches an impressive 93.5%.

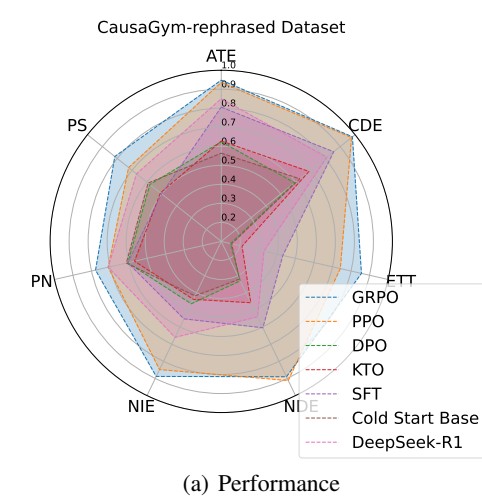

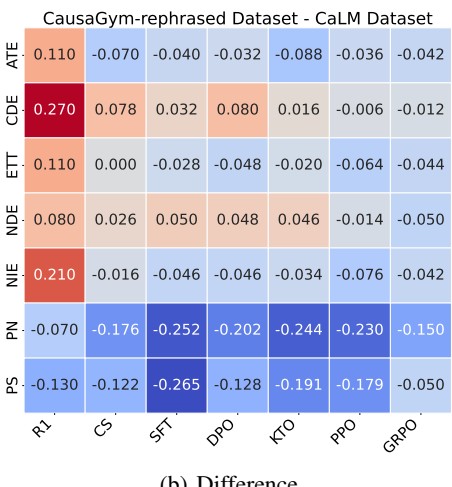

(a) Performance          (b) Difference

Figure 2: (a) Model performance on `CausaGym`-rephrased. (b) Model performance difference between `CausaGym`-rephrased and CaLM. "R1" denotes DeepSeek-R1-0528-671B, "CS" denotes Cold Start Base.

Despite DPO's lower performance compared to other methods, it still enables the *causal inference agent* to perform on a par with DeepSeek-R1-0528-671B. This demonstrates that current advanced post-training methods can significantly enhance the causal inference capabilities of LLMs. (2) On average performance, GRPO proves to be the most effective method for building *causal inference agents*. After training with GRPO, the average performance of the LLM reaches 93.5%. This is 42.5% higher than DeepSeek-R1-Distill-Qwen-14B, and more than 23.3% over SFT. (3) All post-training methods contribute to enhancing the LLM's causal inference abilities to varying degrees. SFT gains 19.2% improvement over the model only with a cold start; DPO gains 1.4% improvement over the model only with a cold start; KTO gains 6.4% improvement over the model only with a cold start; PPO gains 41.1% improvement over the model only with a cold start; GRPO gains 42.5% improvement over the model only with a cold start. (4) In general, online RL methods (GRPO, PPO) demonstrate statistically significant superiority over offline RL methods (DPO, KTO) and SFT. The offline RL post-training methods and SFT gain a 3.9% and 19.8% improvement over the model only with a cold start respectively, while the online methods gain a surprising 41.8% improvement.

### 3.3 GENERALIZATION

Figure 2(a) and Figure 2(b) present the performance of our different training approaches on the `CausaGym`-rephrased and show the performance difference between these approaches on the CaLM dataset and the `CausaGym`-rephrased. We draw the following conclusions: (1) Online RL methods still maintain their superiority on `CausaGym`-rephrased. The average performance of online RL methods GRPO and PPO still reaches 85.7%, while the average performance of offline RL methods and SFT reaches only 48.9% and 62.2%. (2) Post-training methods are robust to paraphrasing. The average performance drop of post-training methods after paraphrasing is only 6.8%, while that of the model only with a cold start is 4.0%. (3) DeepSeek-R1-0528-671B is also robust to paraphrasing. The average of its performance is 65.2%, which is even 8.2% higher than its performance on CaLM.

Table 2 represents the performance of post-training methods on the math test datasets. The performances of all methods are close to the performance of DeepSeek-R1-Distill-Qwen-14B, whose maximum difference is

| LLM | MATH 500 | AMC 2023 | Minvera Math | Avg. |
|---|---|---|---|---|
| DeepSeek-R1-Distill-Qwen-14B | 0.938 | 0.867 | 0.375 | 0.727 |
| SFT | 0.938 | 0.892 | 0.397 | 0.742 |
| DPO | 0.926 | 0.843 | 0.378 | 0.715 |
| KTO | 0.932 | 0.880 | 0.368 | 0.726 |
| PPO | 0.924 | 0.855 | 0.404 | 0.727 |
| GRPO | 0.924 | 0.855 | 0.408 | 0.729 |

Table 2: Performance comparison on three math datasets.

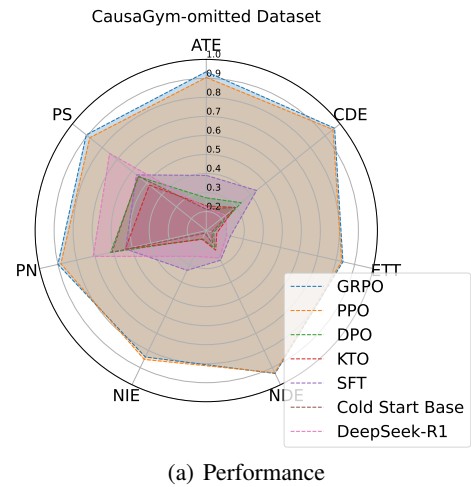

(a) Performance

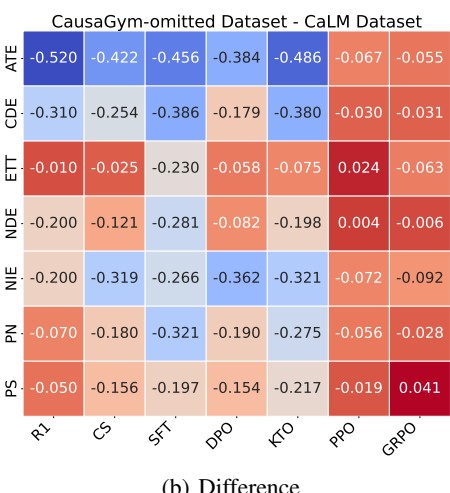

(b) Difference

Figure 3: (a) Model performance on CausaGym-omitted. (b) Model performance difference between CausaGym-omitted and CaLM. "R1" denotes DeepSeek-R1-0528-671B, "CS" denotes Cold Start Base.

less than 2.0%. The result shows that employing post-training on causal inference does not degrade LLM math ability.

## 3.4 INTERNALIZATION

Figure 3(a) and Figure 3(b) present the performance of our different training approaches and DeepSeek-R1-0528-671B on the CausaGym-omitted and show the performance difference between these approaches on the CaLM dataset and the CausaGym-omitted. We find that: (1) Online RL methods still maintain their superiority on CausaGym-omitted. The average performance of online RL methods still reaches 89.6%, while the average performance of offline RL methods and SFT reaches only 30.9% and 39.5%. (2) Online RL methods are robust to removing instructions, but offline RL methods and SFT are not. The average performance drop of online RL after omitting is only 3.2%, while the average performance drop of offline RL and SFT reach 24.0% and 30.7% respectively. In contrast, the average performance drop of the model only with a cold start is 17.1%. (3) DeepSeek-R1-0528-671B is not robust to removing instructions. The average of its performance is 37.5%, which is 19.5% lower than its performance on CaLM.

Figure 4 presents the performance of different training approaches and DeepSeek-R1-0528-671B on the CausaGym-deconfounding. We can conclude that:

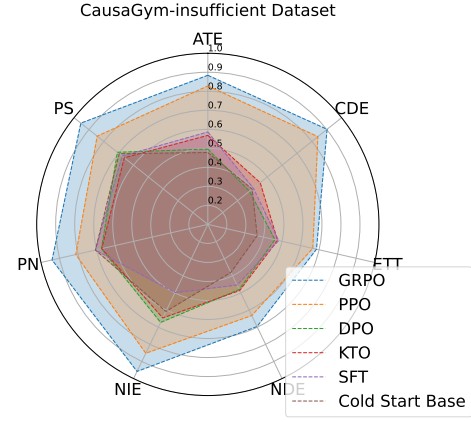

(a) CausaGym-redundant dataset

(b) CausaGym-insufficient dataset

Figure 5: Model performance on CausaGym-redundant dataset and CausaGym-insufficient.

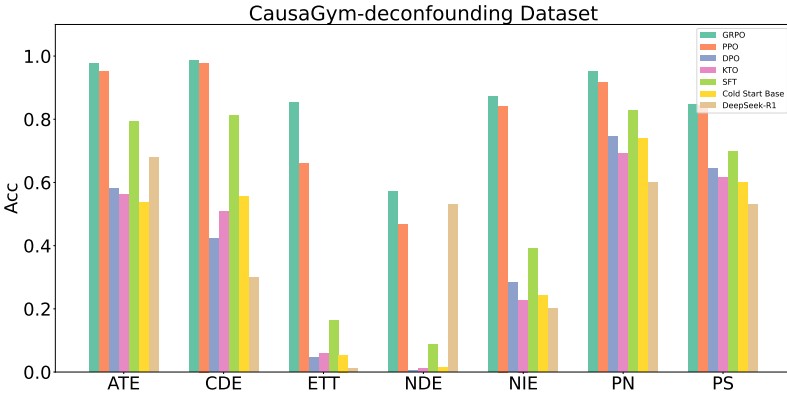

Figure 4: Model performance on CausaGym-deconfounding.

(1) The online RL method GRPO still has the best performance. Its average performance on the CausaGym-deconfounding reaches 86.5%. In contrast, the average performance of KTO, DPO, SFT, PPO is 38.2%, 39.0 %, 53.9% and 80.8%. Moreover, the performance of GRPO is higher than any other post-training method on every casual inference task. (2) Online RL methods perform well on this dataset, while offline RL methods struggle. The average performance of the model with only a cold start is 39.2%. The average improvement of online RL method is 44.4% and that of SFT is 14.7%. However, there is not improvement for offline RL methods. (3) DeepSeek-R1-0528-671B is poor at applying the backdoor criterion. The average performance of DeepSeek-R1-0528-671B is 40.7%, similar to cold start base model.

In general, the result shows that online RL methods, especially GRPO, enables LLM to understand the underlying causal structure of given questions and apply causal inference theorems independently without additional cues. It also reveal that DeepSeek-R1-0528-671B actually struggles at identifying spurious correlations and understanding causal inference tasks.

## 3.5 ROBUSTNESS

Figure 5(a) presents the performance of different training approaches on the CausaGym-redundant Dataset. We draw the following conclusions: (1) The online RL method GRPO still has the best performance. Its average performance on the CausaGym-redundant Dataset reaches 92.0%. In contrast, the average perfor-

mance of KTO, DPO, SFT, PPO is 51.3%, 48.3 %, 66.3% and 88.9%. (2) Online RL methods perform well; SFT has some effect; Offline RL methods work little. The average performance of the model with only a cold start is 50.1%. The average improvement of online RL method is 40.3% and that of SFT is 16.2%. However, there is not improvement for offline RL methods.

Figure 5(b) presents the performance of different training approaches on the `CausaGym`-insufficient Dataset. Our key findings are as follows: (1) The online RL method GRPO still has the best performance. Its average performance on the `CausaGym`-deconfounding reaches 86.5%. In contrast, the average performance of KTO, DPO, SFT, PPO is 56.6%, 55.3%, 54.4% and 78.2%. Moreover, the performance of GRPO is higher than any other post-training method on every casual inference task. (2) Online RL methods perform well on this dataset, while SFT and offline RL methods struggle. The average performance of the model with only a cold start is 51.9%. The average improvement of online RL method is 30.4%, while that of SFT and offline RL methods is about 4.7% and 2.9%, which are negligible. In general, the result shows that online RL methods significantly improve a large language model's ability to identify the appropriate data to solve a given question. Offline RL methods and SFT, however, enhance this ability less.

## 4 RELATED WORKS

**Post-training methods for LLMs.** Since pre-trained LLMs already demonstrate impressive general abilities across a wide range of tasks, recent research focuses on post-training methods to further refine their alignment, reasoning, and problem-solving skills. Post-training typically involves additional supervised tuning or reinforcement learning to align model behavior with human intent, preferences, or reasoning principles. Ouyang et al. (2022) introduce SFT and PPO to align LLMs with human intent, marking the foundation of instruction-following models such as InstructGPT. Building on this line of work, Rafailov et al. (2023) and Ethayarajh et al. (2024) propose DPO and KTO respectively, to more effectively align LLMs with human preferences without requiring explicit reward modeling. Shao et al. (2024) introduce GRPO, an online reinforcement learning method, and demonstrates remarkable success in domains such as mathematical reasoning and code generation.

**LLM causal inference ability.** With the emergence of LLM, researchers have been curious about how well LLMs solve causal inference problems and understand causal concepts. Some research investigates the extent of LLMs' understanding of causality. Zečević et al. (2023) claim that LLMs struggle to memorize and reproduce correlations of causal facts found within their vast training data, while Jin et al. (2024) find that LLMs have the difficulty in determining causal relations from correlation statements. Other works study how LLMs perform on causal tasks, such as interventional and counterfactual problems. Gao et al. (2023) reveal that Chatgpt has a serious hallucination on causal reasoning, making it a bad causal reasoner, while Chen et al. (2024b) present that counterfactual problems, like ETT, NIE, PN, and their numerical version, causal effect estimation are still a challenge to LLMs.

## 5 CONCLUSION

Through comprehensive experiments using the novel `CausaGym` dataset, we demonstrate that post-training can transform a smaller 14B-scale LLM into a highly effective causal inference agent, outperforming larger models. Our research shows that while offline training methods equip LLMs with fundamental causal concepts, online RL methods are crucial for teaching them to apply these rules to solve complex problems. Among the methods tested, GRPO emerges as the most effective, achieving an impressive 93.5% on the CaLM benchmark. This work establishes that online RL, and particularly GRPO, enables LLMs to generalize to rephrased questions, internalize causal theorems without explicit instructions, and robustly handle noisy data, thereby making sophisticated causal inference accessible to a broader audience.

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

# A   THE USE OF LARGE LANGUAGE MODELS

We use a general-purpose LLM in a limited, editorial capacity: to proofread grammar and style, help rephrase a few sentences, and suggest keywords for literature searches. All ideas, analyses, experiments, and writing decisions are our own; the LLM does not generate novel content or influence the study's methodology or results.

# B   PRELIMINARY

## B.1   STRUCTURAL CAUSAL MODEL

Structural Causal Model (SCM) is a way to describe causal-related variables and how they interact with each other (Pearl et al., 2016). An SCM is a triple, represented as $\mathbf{M} = \{\mathbf{U}, \mathbf{V}, \mathbf{F}\}$. $\mathbf{U}$ denotes a set of exogenous variables, whose causes are outside the model. $\mathbf{V}$ denotes a set of endogenous variables, whose values are determined by variables within the model, namely the variables in $\mathbf{V}$ and $\mathbf{U}$. $\mathbf{F}$ denotes a set of functions, which specify how the value of each endogenous variable is determined. Their general form is $X = f_X(PA_X, U_X)$, where $X$ is a variable in $\mathbf{V}$, $U_X$ is a variable in $\mathbf{U}$, and $PA_X$ is a set of variables in $\mathbf{V}$, which have the direct effect to $X$.

An SCM can also be visualized as a directed acyclic graph (DAG) $G$. Its nodes represent the variables in $\mathbf{V}$. There is a directed edge from a variable $Y$ to $X$ in the $G$ if and only if $Y$ is a member of the $PA_X$ set for the function $X = f_X(PA_X, U_X)$.

## B.2   INTERVENTION

Intervention aims to answer the question of "if" ("*If I lower the selling price now, will the sales increase?*"). It can be formally defined with the *do-operator*, $P(Y = y \mid \mathbf{do}(X = x))$. In the SCM, $\mathbf{do}(X = x)$ means replacing the structural equation $X = f_X(PA_X, U_X)$ with $X = x$.

A confounder is a variable that meets all three of these criteria: (1) It is a cause of the outcome, (2) It is associated with the treatment, (3) It is not a consequence of the treatment. If there are confounders for the outcome $Y$ and the treatment $X$, they will create a spurious correlation between the outcome and the treatment, namely $P(Y = y \mid \mathbf{do}(X = x)) \neq P(Y = y \mid X = x)$. To identify the intervention effect with observational data, we can use the Backdoor Criterion.

The backdoor criterion is defined as "Given an ordered pair of variables $(X, Y)$ in a DAG $G$, a set of variables $Z$ satisfies the backdoor criterion relative to $(X, Y)$ if no node in $Z$ is a descendant of $X$, and $Z$ blocks every path between $X$ and $Y$ that contains an arrow into $X$" (Pearl et al., 2016). If a set of variables $Z$ satisfies this criterion for $X$ and $Y$, we can get the following formula:

$$P(Y = y \mid \mathbf{do}(X = x)) = \sum_z P(Y = y \mid X = x, Z = z)P(Z = z). \tag{8}$$

## B.3   COUNTERFACTUAL

Counterfactual addresses the "what-if" question by estimating the values of variables under hypothetical interventions that differ from observed conditions. Formally, a counterfactual problem is often expressed as $P(Y_{X=x} = y \mid X = x', Y = y')$. In this expression, $X = x'$ and $Y = y'$ represent the observed data, while $Y_{X=x}$ denotes the value of Y had the intervention X=x occurred. In a typical counterfactual calculation process, we use these observations to estimate the probability distributions(or the exact values) of exogenous variables. The process typically involves using observed data to estimate the probability distributions of

exogenous variables within a causal model. Subsequently, an intervention is applied to the SCM, such as $\mathbf{do}(X = x)$, to derive the final counterfactual outcomes. The connection between counterfactual inference and intervention inference is that $P(Y = y \mid \mathbf{do}(X = x), Z = z) = P(Y_{X=x} = y \mid Z_{X=x} = z)$.

To identify and compute the counterfactual probability $P(Y_{X=x} = y)$ directly from empirical, we can employ a theorem known as the Counterfactual Interpretation of Backdoor (Pearl et al., 2016). It states that if a set of variables, $Z$, satisfies the backdoor condition with respect to the causal relationship from $X$ to $Y$, then for any value $x$, the counterfactual outcome $Y_{X=x}$ is conditionally independent of the actual treatment $X$ given $Z$. This key property is formally expressed as:

$$P(Y_{X=x} = y \mid X = x', Z = z) = P(Y_{X=x} = y \mid Z = z). \tag{9}$$

### B.4 CAUSAL INFERENCE TASKS

In this paper, we delineate the following seven key causal inference tasks (Pearl, 2009; Pearl & Mackenzie, 2018): (1) **Average Treatment Effect (ATE).** The expected difference in outcomes had everyone received treatment versus had everyone received no treatment. (2) **Controlled Direct Effect (CDE).** The expected difference in outcomes had everyone received treatment versus had everyone received no treatment, while holding the mediator variable at a specific level. (3) **Effect of the Treatment on the Treated (ETT).** The expected difference in outcomes for the subpopulation that actually received the treatment. (4) **Natural Direct Effect (NDE).** The effect of the treatment on the outcome, with the mediator set to the value it would naturally take in the absence of treatment. (5) **Natural Indirect Effect (NIE).** The effect on the outcome transmitted solely through the mediator, when the treatment is changed from no treatment to treatment. (6) **Probability of Necessity (PN).** The probability that the absence of treatment was a necessary condition for the outcome to be absent, given that treatment was received and the outcome occurred. (7) **Probability of Sufficiency (PS).** The probability that receiving treatment was a sufficient condition for the outcome to occur, given that no treatment was received and the outcome did not occur.

## C DATASET GENERATION TEMPLATE

Table 3 lists all templates used for generating questions of different causal tasks.

## D TESTING DATASET EXAMPLES

Figure 6, 7, 8, 9 and 10 provide examples of `CausaGym` test sets. They exhibit features of each dataset.

## E OTHER MODEL PERFORMANCE ON CAUSAGYM

### E.1 BASE MODEL VARIANTS

To further demonstrate the advantage of online RL methods in improving the causal inference capability of LLMs, we further train and evaluate multiple base models, including Mistral-7B and DeepSeek-R1-Distill-Llama-8B. We apply GRPO, DPO, and SFT to these two models, and the result is shown in Table 4. The strong performance of GRPO models further confirms that its effects consistently extend beyond a single model.

### E.2 GRPO VARIANTS

**CausaGym-rephrased**

Modified Problem:
Input Info: Consider an isolated hypothetical scenario governed solely by these specified relationships, with no additional influencing factors: The level of traffic congestion directly influences appearance. Traffic congestion also directly impacts rainfall. Additionally, appearance has a direct causal effect on temperature.
When traffic congestion is low, the likelihood of appearance being high is 0.4358. Conversely, when traffic congestion is high, the probability of appearance being high drops to 0.0511.

Instruction: Evaluate the effect of treatment on the treated (ETT) regarding how traffic congestion affects appearance.

Query: For individuals currently experiencing low traffic congestion, had their congestion level instead been high, would their probability of having high appearance be reduced?
Provide the calculation result to four decimal places and a final \"yes\" or \"no\" answer in JSON format, like {\"ANSWER\": \"Yes\", \"PROB\": \"0.1234\"}

Original Problem:
Input Info: Imagine a self-contained, hypothetical world with only the following conditions, and without any unmentioned factors or causal relationships: Traffic congestion has a direct effect on appearance. Traffic congestion has a direct effect on rainfull. Appearance has a direct effect on temperature.
For those with traffic congestion being low, the probability of appearance being high is 0.4358. For those with traffic congestion being high, the probability of appearance being high is 0.0511.

Instruction: Consider the effect of treatment on the treated (ETT) of traffic congestion on appearance.

Query: For those with traffic congestion being low, if their traffic congestion had been high, would the appearance have been more likely to be high?
Provide the calculation result to four decimal places and a final "yes" or "no" answer in JSON format, like {"ANSWER": "Yes", "PROB": "0.1234"}:

Figure 6: An example for `CausaGym`-rephrased. The input info, instruction, and query are rephrased to test the robustness and overfitting tendency of the LLMs.

**CausaGym-omitted**

Modified Problem:
Input Info: Consider an isolated, hypothetical scenario governed solely by the following relationships, with no additional influencing factors: The level of traffic congestion directly influences appearance. Traffic congestion also directly affects rainfull. Additionally, appearance has a direct impact on temperature.
For individuals experiencing low traffic congestion, the likelihood of appearance being high is 0.4358. Conversely, for those with high traffic congestion, the probability of appearance being high is 0.0511.

Query: Among those currently with low traffic congestion, had their traffic congestion instead been high, would the probability of appearance being high be lower?\nProvide the calculation result to four decimal places and a final \"yes\" or \"no\" answer in JSON format, like {\"ANSWER\": \"Yes\", \"PROB\": \"0.1234\"}:"

Original Problem:
Input Info: Imagine a self-contained, hypothetical world with only the following conditions, and without any unmentioned factors or causal relationships: Traffic congestion has a direct effect on appearance. Traffic congestion has a direct effect on rainfull. Appearance has a direct effect on temperature.
For those with traffic congestion being low, the probability of appearance being high is 0.4358. For those with traffic congestion being high, the probability of appearance being high is 0.0511.

Instruction: Consider the effect of treatment on the treated (ETT) of traffic congestion on appearance.

Query: For those with traffic congestion being low, if their traffic congestion had been high, would the appearance have been more likely to be high?
Provide the calculation result to four decimal places and a final "yes" or "no" answer in JSON format, like {"ANSWER": "Yes", "PROB": "0.1234"}:

Figure 7: An example for `CausaGym`-omitted. The input info and query are rephrased while the instruction is omitted, in order to assess whether the LLMs can correctly recognize the underlying causal tasks in the problem.

**CausaGym-deconfounding**

Problem：

Input Info:Imagine a self-contained, hypothetical world with only the following conditions, and without any unmentioned factors or causal relationships:Talent has a direct effect on market demand.Talent has a direct effect on wheather condition.Wheather condition has a direct effect on stress level.Market demand has a direct effect on government policies.Market demand has a direct effect on work-life balance.Market demand has a direct effect on physical health.Market demand has a direct effect on stress level.Physical health has a direct effect on work-life balance.Physical health has a direct effect on stress level.Physical health has a direct effect on rainfall.Work-life balance has a direct effect on stress level.Work-life balance has a direct effect on rainfall.Stress level has a direct effect on rainfall.Rainfall has a direct effect on government policies.

For those with market demand being low (C=0), work-life balance being low (E=0), physical health being high (D=1) and talent being low (A=0), the probability of stress level being low (F=0) is 0.7671. For those with market demand being high (C=1), work-life balance being low (E=0), physical health being high (D=1) and talent being low (A=0), the probability of stress level being low (F=0) is 0.8770. The probability of talent being low (A=0) is 0.6188. For those with market demand being low (C=0), work-life balance being low (E=0), physical health being high (D=1) and talent being high (A=1), the probability of stress level being low (F=0) is 0.7114. For those with market demand being high (C=1), work-life balance being low (E=0), physical health being high (D=1) and talent being high (A=1), the probability of stress level being low (F=0) is 0.8610. The probability of talent being high (A=1) is 0.3812.

Instruction: Consider the controlled direct effect (CDE) of market demand on stress level.

Query: Conditioned on work-life balance being low and physical health being high, if the market demand had been low, would the stress level have been more likely to be low? Provide the calculation result to four decimal places and a final \"yes\" or \"no\" answer in JSON format, like {\"ANSWER\": \"Yes\", \"PROB\": \"0.1234\"}:

Figure 8: An example for `CausaGym`-deconfounding. "Talent" functions as a confounder in this problem. To remove its spurious influence, the backdoor criterion needs to be applied.

**CausaGym-redundant**

Modified Problem：

Input Info: Imagine a self-contained, hypothetical world with only the following conditions, and without any unmentioned factors or causal relationships:Jqrv has a direct effect on noja.Noja has a direct effect on axcm.Noja has a direct effect on ddzl.Noja has a direct effect on gtyu.Axcm has a direct effect on chfs.Lnqx has a direct effect on gtyu.Lnqx has a direct effect on ddzl.Gtyu has a direct effect on ddzl.Chfs has a direct effect on gitz.

For those with chfs being low (F=0) and axcm being low (C=0), the probability of gitz being low (H=0) is 0.1300. For those with axcm being high (C=1), the probability of chfs being low (F=0) is 0.3295. For those with axcm being low (C=0), the probability of chfs being low (F=0) is 0.6173. For those with chfs being high (F=1) and axcm being low (C=0), the probability of gitz being low (H=0) is 0.0134. For those with noja being low (B=0), the probability of ddzl being low (G=0) is 0.5750. For those with gtyu being high (E=1), lnqx being low (D=0) and noja being low (B=0), the probability of ddzl being high (G=1) is 0.5954.

Instruction: Consider the natural indirect effect (NIE) of axcm on gitz.

Query: Suppose axcm is held constant and the mediator changes to whatever value it would have attained under axcm changing to be high, would gitz have been more likely to be low?\nProvide the calculation result to four decimal places and a final \"yes\" or \"no\" answer in JSON format, like {\"ANSWER\": \"Yes\", \"PROB\": \"0.1234\"}:

Original Problem：

Input Info: Imagine a self-contained, hypothetical world with only the following conditions, and without any unmentioned factors or causal relationships:Jqrv has a direct effect on noja.Noja has a direct effect on axcm.Noja has a direct effect on ddzl.Noja has a direct effect on gtyu.Axcm has a direct effect on chfs.Lnqx has a direct effect on gtyu.Lnqx has a direct effect on ddzl.Gtyu has a direct effect on ddzl.Chfs has a direct effect on gitz.

For those with chfs being low (F=0) and axcm being low (C=0), the probability of gitz being low (H=0) is 0.1300. For those with axcm being high (C=1), the probability of chfs being low (F=0) is 0.3295. For those with axcm being low (C=0), the probability of chfs being low (F=0) is 0.6173.

Instruction: Consider the natural indirect effect (NIE) of axcm on gitz.

Query: Suppose axcm is held constant and the mediator changes to whatever value it would have attained under axcm changing to be high, would gitz have been more likely to be low?\nProvide the calculation result to four decimal places and a final \"yes\" or \"no\" answer in JSON format, like {\"ANSWER\": \"Yes\", \"PROB\": \"0.1234\"}:

Figure 9: An example for `CausaGym`-redundant. Irrelevant conditions are introduced into the input info to test whether the LLMs remain robust against irrelevant interference.

**CausaGym-insufficient**

**Modified Problem:**

Input Info: Imagine a self-contained, hypothetical world with only the following conditions, and without any unmentioned factors or causal relationships:Student's study habits has a direct effect on student's grades.Student's attendance has a direct effect on student's participation in extracurricular activities.Student's attendance has a direct effect on student's motivation.Student's grades has a direct effect on student's academic performance.Student's grades has a direct effect on student's motivation.Student's academic performance has a direct effect on student's college admission.Student's participation in extracurricular activities has a direct effect on student's leadership skills.

For those with student's attendance being excellent (B=1), the probability of student's leadership skills being strong (G=1) is 0.6751.

Instruction: Consider the probability of sufficiency (PS) of student's attendance on student's leadership skills.

Query: Given that student's attendance was poor and student's leadership skills was lacking, what is the upper bound of the probability that student's leadership skills would have been strong if the student's attendance had been excellent?\nProvide the calculation result to four decimal places in JSON format, like {\"PROB\": \"0.1234\"}:

**Original Problem:**

Input Info: Imagine a self-contained, hypothetical world with only the following conditions, and without any unmentioned factors or causal relationships:Student's study habits has a direct effect on student's grades.Student's attendance has a direct effect on student's participation in extracurricular activities.Student's attendance has a direct effect on student's motivation.Student's grades has a direct effect on student's academic performance.Student's grades has a direct effect on student's motivation.Student's academic performance has a direct effect on student's college admission.Student's participation in extracurricular activities has a direct effect on student's leadership skills.

For those with student's attendance being excellent (B=1), the probability of student's leadership skills being strong (G=1) is 0.6751. The probability of student's attendance being poor (B=0) and student's leadership skills being lacking (G=0) is 0.1426. The probability of student's attendance being excellent (B=1) and student's leadership skills being strong (G=1) is 0.4349.

Instruction: Consider the probability of sufficiency (PS) of student's attendance on student's leadership skills.

Query: Given that student's attendance was poor and student's leadership skills was lacking, what is the upper bound of the probability that student's leadership skills would have been strong if the student's attendance had been excellent?\nProvide the calculation result to four decimal places in JSON format, like {\"PROB\": \"0.1234\"}:

Figure 10: An example for `CausaGym`-insufficient. Necessary conditional probabilities are omitted from the input info to assess whether the LLMs can detect the missing information required for correct causal inference.

| Causal Tasks | Template |
|---|---|
| ATE | If {{treatment}} is changed to be {{treatment_value}}, will the {{outcome}} be more likely to be {{outcome_value}}? |
| ETT | For those with {{treatment}} being {{treatment_value}}, if their {{treatment}} had been {{not_treatment_value}}, would {{outcome}} have been more likely to be {{outcome_value}}? |
| CDE | Conditioned on {{mediator_1}} being {{mediator_1_value}}, {{mediator_2}} being {{mediator_2_value}}, …, {{mediator_n}} being {{mediator_n_value}}, if {{treatment}} had been {{treatment_value}}, would {{outcome}} have been more likely to be {{outcome_value}}? |
| NIE | Suppose {{treatment}} is held constant and the mediator changes to whatever value it would have attained under {{treatment}} changing to be {{treatment_value}}, would the {{outcome}} have been more likely to be {{outcome_value}}? |
| NDE | Suppose the mediator keeps constant when {{treatment}} is changed to be {{treatment_value}}, would the {{outcome}} have been more likely to be {{outcome_value}}? |
| PS | Given that {{treatment}} was {{treatment_negative}} and {{outcome}} was {{outcome_negative}}, what is the lower bound and upper bound of the probability that {{outcome}} would have been {{outcome_positive}} if the {{treatment}} had been {{treatment_positive}}? |
| PN | Given that {{treatment}} was {{treatment_positive}} and {{outcome}} was {{outcome_positive}}, what is the lower bound and upper bound of the probability that {{outcome}} would have been {{outcome_negative}} if the {{treatment}} had been {{treatment_negative}}? |

Table 3: Question templates for different causal tasks.

| Base Model | Method | CaLM | Omitted | Redundant | Rephrased |
|---|---|---|---|---|---|
| Mistral-7B | Base | 0.151 | 0.117 | 0.124 | 0.163 |
| | DPO | 0.067 | 0.038 | 0.048 | 0.048 |
| | SFT | 0.411 | 0.266 | **0.421** | 0.374 |
| | GRPO | **0.511** | **0.479** | 0.363 | **0.479** |
| DeepSeek-R1-Distill-Llama-8B | Base | 0.213 | 0.152 | 0.136 | 0.256 |
| | DPO | 0.357 | 0.260 | 0.311 | 0.319 |
| | SFT | 0.560 | 0.315 | 0.525 | 0.490 |
| | GRPO | **0.861** | **0.843** | **0.829** | **0.811** |

Table 4: Performance comparison across different base LLMs and post-training methods

Table 5 shows the performance of two GRPO variants on `CausaGym` test sets. GRPO-no-think is a GRPO model trained without the reasoning format reward, while all other training configurations remain identical to the GRPO model described in the main text. Its performance exhibits only a marginal decline relative to the original GRPO model, indicating that the strong overall results primarily stem from the GRPO training framework itself rather than the particular reward design. Realistic-GRPO is trained with the same training procedure described in the main text, but the training data include only those questions in which the variable names are semantically meaningful. The significant decrease in performance suggests that the introduction of random symbolic variables induces beneficial variability, mitigating reliance on superficial semantic cues and thereby facilitating improved robustness, internalization and generalization.

| Model | CaLM | Deconfounding | Insufficient | Omitted | Redundant | Rephrased |
|-------|------|---------------|--------------|---------|-----------|-----------|
| GRPO-no-think | 0.941 | 0.834 | 0.891 | 0.651 | 0.905 | 0.868 |
| Realistic-GRPO | 0.891 | 0.763 | 0.887 | 0.527 | 0.879 | 0.760 |

Table 5: Performance comparison of GRPO variants on CaLM and `CausaGym` test sets.

## F COMPARISON WITH OTHER CAUSAL REASONING BENCHMARK

To show the difference between our benchmark `CausaGym` and other causal reasoning benchmarks, we compare `CausaGym` with CLadder (Jin et al., 2023), CaLM (Chen et al., 2024b) and CLEAR (Chen et al., 2024c) and the result is shown in Table 6.

| Benchmark | Training Dataset | Numerical Input | Rationale | Generalization Test | Internalization Test | Robustness Test |
|-----------|------------------|-----------------|-----------|---------------------|----------------------|-----------------|
| CLadder | ✗ | ✔ | ✔ | ✗ | ✗ | ✗ |
| CALM | ✗ | ✔ | ✔ | ✗ | ✗ | ✗ |
| CLEAR | ✗ | ✗ | ✗ | ✗ | ✗ | ✗ |
| CausaGym | ✔ | ✔ | ✔ | ✔ | ✔ | ✔ |

Table 6: Comparison of causal reasoning benchmarks.

## G TRAINING DATASET STATISTICS

To demonstrate the question diversity and entity variety of our training dataset, we list its critical characteristics in Table 7, where probability count refers to the number of distinct probabilities that appear in each question.

.

| Maximal Node Number | Maximal Probability Count | Average Probability Count | Question Type | Sample |
|---------------------|---------------------------|---------------------------|---------------|--------|
| 10 | 12 | 3.84 | 7 | 17500 |

Table 7: Characteristics of the `CausaGym` training dataset.

## H GRPO MODEL PERFORMANCE ON OTHER CAUSAL BENCHMARK

To further demonstrate the generalization capability of GRPO model, we evaluate the performance of GRPO model and its base model, DeepSeek-R1-Distill-Qwen-14B on other causal reasoning benchmarks, including CLadder, CaLM, CLEAR and CausalProbe-E (Chi et al., 2024). As shown in Table 8, the result demonstrates impressive generalization capability of GRPO model.

| Benchmark | GRPO | Base | Best Performance Reported in the Original Paper |
|-----------|------|------|-------------------------------------------------|
| CaLM (Mathematical) | 0.935 | 0.272 | 0.270 (GPT-3.5-Turbo) |
| CLadder | 0.719 | 0.577 | 0.704 (GPT-4 + CausalCOT) |
| CLEAR | 0.472 | 0.433 | 0.605 (GPT-4) |
| CausalProbe-E | 0.805 | 0.803 | 0.750 (Claude 3.5 Opus) |

Table 8: Performance on other causal reasoning benchmarks.

# I   POST-TRAINED MODEL'S REASONING PROCESS

Figure 11 demonstrates an example of how the GRPO model reason.

---

**Model's reasoning process**

<think>
...
In this context, the treatment is amount of exercise (C), and the outcome is sales performance (H). The mediator is stress level (F), I think, because it's mentioned that stress level affects sales performance and is affected by exercise.   ✔ *Correct formula and explanation*
...
In this case, since B is a confounder that affects both C and F, and F and H, we need to condition on B.
So, NIE = sum_{B} P(B) * [ sum_{F} P(H=0 | C=1, F, B) * (P(F | C=0, B) - P(F | C=1, B)) ]
Yes, this makes sense. Because for each B, we have the indirect effect when C=1 but F as if C=0, and then we average over B. In the expression, P(F | C=0, B) - P(F | C=1, B) is the change in F distribution when C changes from 1 to 0, for fixed B. Then P(H=0 | C=1, F, B) is the probability of H=0 when C=1 and F is at that value, which is what it would be if F were as in C=0. This captures the indirect effect.
...
Let me add: 0.002918 (from B=0) and 0.018936 (from B=1) = 0.021854, so 0.0219 when rounded to four decimals. So, for the output, it should be {"ANSWER": "Yes", "PROB": "0.0219"}.
</think>

The natural indirect effect (NIE) of amount of exercise (C) on sales performance (H) through the mediator stress level (F) is calculated using the given probabilities and confounder market demand (B). The NIE is computed as follows:

- For B=0:

$$\text{NIE}_{B=0} = \sum_F P(H = 0 \mid C = 1, F, B = 0) \cdot [P(F \mid C = 0, B = 0) - P(F \mid C = 1, B = 0)]$$
$$= \cdots = 0.9541 \cdot (0.1141 - 0.5758) + 0.9922 \cdot (0.8859 - 0.4242) \approx 0.0176$$

- For B=1:

$$\text{NIE}_{B=1} = \sum_F P(H = 0 \mid C = 1, F, B = 1) \cdot [P(F \mid C = 0, B = 1) - P(F \mid C = 1, B = 1)]$$
$$= \cdots = 0.8944 \cdot (0.0476 - 0.3509) + 0.9693 \cdot (0.9524 - 0.6491) \approx 0.0227$$

- Overall NIE:

$$\text{NIE} = P(B = 0) \cdot \text{NIE}_{B=0} + P(B = 1) \cdot \text{NIE}_{B=1} = 0.1658 \cdot 0.0176 + 0.8342 \cdot 0.0227 \approx 0.021869$$
Rounded to four decimal places: 0.0219

{"ANSWER": "Yes", "PROB": "0.0219"}   ✔ *Formatted output*

---

Figure 11: An example for GRPO model's reasoning process.

