# OpenReview forum: "Can LLMs Serve as Causal Inference Agents? A Study on Post-Training Methods"
_ICLR.cc/2026/Conference — ICLR 2026 Conference Withdrawn Submission_

### Official Review · Reviewer_srnV · 2025-10-26

**Soundness:** 3
**Presentation:** 3
**Contribution:** 2
**Rating:** 4
**Confidence:** 4

**Summary:**

The author introduced the DeepCausa dataset (with variants) for training large language models in causal reasoning and evaluated the effectiveness of popular post-training methods in enabling models to perform this task.

**Strengths:**

1. The work explores the suitability of using LLMs for certain causal tasks, representing a non-trivial effort that provides valuable insights and lays the groundwork for further research.
2. The experiments are comprehensive and well-documented, offering detailed analyses across multiple aspects, including generalization, internalization, and robustness of the models.

**Weaknesses:**

1. The causal tasks and datasets used in this paper assume that the required statistics (e.g., probabilities for do-operators) are explicitly provided in the questions, meaning the LLM’s role is largely retrieving numbers and performing simple arithmetic. This assumption limits the applicability of the framework to more complex causal reasoning tasks.
2. In the motivation example in Figure 1, only two probabilities are provided: P(B=1|A=0) and P(B=1|A=1). Probabilities like P(B=0|A=1) or P(B=0|A=0) are omitted, which seems sufficient for computation. If all necessary information is provided in this way, the problem could be solved heuristically without requiring an LLM, raising questions about the necessity of the model.
3. One of the main contributions of this work is the training dataset. However, without a clearer understanding of its quality (such as question diversity and entity variety), its usefulness for further research is limited. Providing supplementary material with detailed dataset statistics would strengthen the contribution.

**Questions:**

N/A

---

> ### Author Response · Authors · 2025-12-02
> **Response (1/1)**
>
> > Weakness 1: The causal tasks and datasets used in this paper assume that the required statistics (e.g., probabilities for do-operators) are explicitly provided in the questions, meaning the LLM’s role is largely retrieving numbers and performing simple arithmetic. This assumption limits the applicability of the framework to more complex causal reasoning tasks.
>
> A1: Our benchmark already includes tasks where key statistics are either missing (DeepCausa-insufficient) or redundant (DeepCausa-redundant) and tasks where the backdoor adjustment is always necessary (DeepCausa-deconfounding), and the GRPO-trained model maintains comparable performance under these more complex conditions. The examples of these test sets are shown in Appendix D.
>
> It achieves accuracy of 86.5% on DeepCausa-insufficient, 92.0% on DeepCausa-redundant and 86.5%, which are much higher than the performance of the cold start base model on these test sets. These results show that our framework does not merely rely on numerical retrieval but can perform robust causal reasoning even when quantitative information is incomplete or noisy.
>
> >Weakness 2: In the motivation example in Figure 1, only two probabilities are provided: P(B=1|A=0) and P(B=1|A=1). Probabilities like P(B=0|A=1) or P(B=0|A=0) are omitted, which seems sufficient for computation. If all necessary information is provided in this way, the problem could be solved heuristically without requiring an LLM, raising questions about the necessity of the model.
>
> A2: Figure 1 simply presents a shorter example for illustration, while the full testing sets contain more complex causal structures and richer probability information.
>
> In DeepCausa‑deconfounding, each question involves at least five variables and includes no fewer than three probability conditions, which makes pure arithmetic solutions infeasible. Moreover, in DeepCausa‑redundant dataset, each question additionally contains two extra redundant probability conditions, testing the model’s ability to identify relevant information under noisy inputs. The examples of DeepCausa-deconfounding and DeepCuasa-redundant are shown in Appendix D. These design choices ensure that the benchmark tests genuine causal reasoning ability rather than simple computational heuristics.
>
> > Weakness 3: One of the main contributions of this work is the training dataset. However, without a clearer understanding of its quality (such as question diversity and entity variety), its usefulness for further research is limited. Providing supplementary material with detailed dataset statistics would strengthen the contribution.
>
> A3: We list key dataset statistics in Table R3. The main characteristic of our training dataset is that it includes complex numerical causal reasoning problems, where probability count refers to the number of distinct probabilities that appear in each
> question.
>
> **Table R3**:  DeepCausa training dataset statistics
>
> | **Maximal Node Number** | **Maximal Probability Count** | **Average Probability Count** | **Question Type** | **Sample** |
> | :-----------------------: | :---------------------------------: | :-----------------------------: | :-----------------: | :----------: |
> | 10                      | 12                                | 3.84                          | 7                 | 17500      |
>
> As a final note, although our benchmark has been renamed CausaGym, we refer to it as DeepCausa in the rebuttal to maintain consistency with the reviewers’ comments.

---

### Official Review · Reviewer_cKjQ · 2025-10-26

**Soundness:** 3
**Presentation:** 2
**Contribution:** 2
**Rating:** 4
**Confidence:** 5

**Summary:**

This paper studies the influence of post-training techniques on the LLMs' causal inference abilities. Authors introduce the DeepCausa benchmark, a comprehensive dataset that contains seven core training causal tasks and five test tasks. In the experiments, the authors evaluate five post-training methods, including SFT, DPO, KTO, PPO, and GRPO. Finally, authors conclude that post-training approaches can enhance LLMs' causal inference abilities.

**Strengths:**

1. The studied problem is important and practical. Both post-training methods and causal reasoning are essential for LLMs.

2. The results analyses are comprehensive, and the authors analyze diverse aspects of agents, including the generalization, internalization, and robustness.

**Weaknesses:**

1. The proposed benchmark and source codes are not open-sourced.

2. I think the motivation may be a little contradictory. Specifically, in the introduction, the authors claim that "we can develop a causal
inference agent that explains its assumptions and reasoning in plain language." However, in Figure 1, it seems that LLMs are still mainly doing the numerical calculation rather than a detailed language explanation. Besides, the term "backdoor adjustment set" may still be hard to understand for non-experts in causal inference.

3. Since there already exist other formal causal reasoning benchmarks (e.g., the CLADDER [1]), I would suggest that authors add an individual section on the differences between their newly proposed DeepCausa and existing benchmarks. Why can‘t other benchmarks test the abilities of post-training methods?

4. The related work section is too short. I think there should be at least two separate parts: post-training methods for LLMs, and LLMs' causal inference abilities. Authors should consider revising the related work.

5. I think Table 1 is unclear and could be misleading. Authors should consider listing other models, post-training methods, and base model (DeepSeek-R1-Distill-Qwen-14B) with separate lines. Currently, it's hard for readers to tell which one is the baseline simply from the table.

6. I believe only one base model (DeepSeek-R1-Distill-Qwen-14B) is not enough. Authors should consider including more base models to verify the generality of their findings.

> [1] Jin Z, Chen Y, Leeb F, et al. Cladder: Assessing causal reasoning in language models[J]. Advances in Neural Information Processing Systems, 2023, 36: 31038-31065.

**Questions:**

Please refer to the weaknesses part.

---

> ### Author Response · Authors · 2025-12-02
> **Response (1/2)**
>
> > Weakness 1: The proposed benchmark and source codes are not open-sourced.
>
> A1: We have updated our dataset and code at https://anonymous.4open.science/r/DeepCausa-137C.
>
> > Weakness 2: I think the motivation may be a little contradictory. Specifically, in the introduction, the authors claim that "we can develop a causal inference agent that explains its assumptions and reasoning in plain language." However, in Figure 1, it seems that LLMs are still mainly doing the numerical calculation rather than a detailed language explanation. Besides, the term "backdoor adjustment set" may still be hard to understand for non-experts in causal inference.
>
> A2: Thank for your suggestions. We apologize for the ambiguity, but the step-by-step symbolic solution in Figure 1 is not an LLM reasoning process. They are only used for generating detail reasoning processes for offline post-training. An example of how LLM reason is shown in Appendix I. In addition, even if the LLM uses terms that are unfamiliar to non-experts, such as 'backdoor adjustment set,' users can interact with the LLM to seek further explanations.
>
> > Weakness 3: Since there already exist other formal causal reasoning benchmarks (e.g., the CLADDER [1]), I would suggest that authors add an individual section on the differences between their newly proposed DeepCausa and existing benchmarks. Why can‘t other benchmarks test the abilities of post-training methods?
>
> A3: We compare DeepCausa and existing causal reasoning benchmarks in Table R1. Unlike existing benchmarks such as CLadder, CaLM, and CLEAR, our DeepCausa benchmark uniquely provides both training and evaluation data, which makes it more suitable for comparing different post-training methods.
>
> Existing causal reasoning benchmarks mainly serve as evaluation-only resources. While they assess model reasoning, they do not include paired training datasets constructed under the same causal framework. As a result, these benchmarks cannot be directly used to compare different post-training methods like GRPO and SFT. DeepCausa addresses this gap by providing an integrated dataset for both training and evaluation, enabling consistent assessment of causal reasoning ability before and after post-training.
>
> **Table R1**: Comparison between DeepCausa and existing causal reasoning benchmarks:
>
> | **Benchmark** | **Training Dataset** | **Numerical Input** | **Rationale** | **Generalization Test** | **Internalization Test** | **Robustness Test** |
> | ------------- | -------------------- | ------------------- | ------------- | ----------------------- | ------------------------ | ---------------------- |
> | CLadder       | No                   | Yes                 | Yes           | No                      | No                       | No                     |
> | CaLM          | No                   | Yes                 | Yes           | No                      | No                       | No                     |
> | CLEAR         | No                   | No                  | No            | No                      | No                       | No                     |
> | DeepCausa     | Yes                  | Yes                 | Yes           | Yes                     | Yes                      | Yes                    |

---

> > ### Author Response · Authors · 2025-12-02
> > **Response (2/2)**
> >
> > >Weakness 4: The related work section is too short. I think there should be at least two separate parts: post-training methods for LLMs, and LLMs' causal inference abilities. Authors should consider revising the related work.
> > > Weakness 5: I think Table 1 is unclear and could be misleading. Authors should consider listing other models, post-training methods, and base model (DeepSeek-R1-Distill-Qwen-14B) with separate lines. Currently, it's hard for readers to tell which one is the baseline simply from the table.
> >
> > A4: Thank you for your suggestions. We have updated the related work and Table 1.
> >
> > > Weakness 6: I believe only one base model (DeepSeek-R1-Distill-Qwen-14B) is not enough. Authors should consider including more base models to verify the generality of their findings.
> >
> > A5: We further train and evaluate multiple base models, including Mistral-7B and DeepSeek-R1-Distill-Llama-8B, to confirm that the effects of GRPO consistently extend beyond a single model.
> >
> > To examine whether our conclusions hold across architectures, we apply GRPO, DPO, and SFT to both Mistral-7B and DeepSeek-R1-Distill-Llama-8B. Then, we evaluate all models on the generalization (DeepCausa-rephrased), internalization (DeepCausa-omitted), and robustness (DeepCausa-redundant) datasets, and the result is shown in Table R2.
> >
> > The consistent improvements achieved by GRPO across both models demonstrate that the advantages of online RL training are not specific to a single base model but generalize to models of different architectures and scales.
> >
> > **Table R2**: Comparison of post-training methods across multiple base models:
> >
> > | **Base Model**               | **Post-training** | **CaLM**  | **Omitted** | **Redundant** | **Rephrased** |
> > | ---------------------------- | ----------------- | --------- | ----------- | ------------- | ------------- |
> > | Mistral-7B                   | Base              | 0.151     | 0.117       | 0.124         | 0.163         |
> > |                              | DPO               | 0.067     | 0.038       | 0.048         | 0.048         |
> > |                              | SFT               | 0.411     | 0.266       | **0.421**     | 0.374         |
> > |                              | GRPO              | **0.511** | **0.479**   | 0.363         | **0.479**     |
> > | DeepSeek-R1-Distill-Llama-8B | Base              | 0.213     | 0.152       | 0.136         | 0.256         |
> > |                              | DPO               | 0.357     | 0.260       | 0.311         | 0.319         |
> > |                              | SFT               | 0.560     | 0.315       | 0.525         | 0.490         |
> > |                              | GRPO              | **0.861** | **0.843**   | **0.829**     | **0.811**     |
> >
> > As a final note, although our benchmark has been renamed CausaGym, we refer to it as DeepCausa in the rebuttal to maintain consistency with the reviewers’ comments.

---

### Official Review · Reviewer_3eeJ · 2025-10-28

**Soundness:** 2
**Presentation:** 3
**Contribution:** 2
**Rating:** 4
**Confidence:** 4

**Summary:**

This paper introduces **DeepCausa**, a new benchmark and dataset for evaluating and training LLMs on causal inference tasks. It formulates seven core causal estimation problems (ATE, CDE, ETT, NDE, NIE, PN, PS) as natural-language reasoning questions, and enables reinforcement learning (especially GRPO) by providing programmatically computable rewards. Experiments show that a 14B model trained with GRPO reaches ~93% accuracy on the benchmark, surpassing larger models, while maintaining performance on math reasoning tasks.

**Strengths:**

1. Introduces the first causal dataset that supports RL-based training with automatic rewards.
2. Provides quantitative evidence that GRPO can improve causal reasoning accuracy without degrading general reasoning (math).

**Weaknesses:**

1. The “agent” claim is overstated; the model remains a passive CoT generator without environment interaction or intervention ability.
2. Evaluation is confined to the same synthetic distribution used for training; no results on external causal benchmarks (CLadder, CLEAR, CaLM) are reported.
3. No ablation on reward shaping or robustness to real data.

**Questions:**

1. How well would the trained model transfer to unseen causal benchmarks such as CaLM or CLadder?
2. Could the benchmark be extended to allow environment-level interaction (e.g., tool-based causal discovery)?
3. Does the model’s performance degrade when SCM variables have realistic semantics rather than random symbols?

---

> ### Author Response · Authors · 2025-12-02
> **Response (1/2)**
>
> >Weakness 1: The “agent” claim is overstated; the model remains a passive CoT generator without environment interaction or intervention ability.
>
> A1: We adopt the term "agent" following prior work to emphasize the model's active reasoning and causal understanding, rather than implying full environment-level interaction or intervention capability. In line with previous studies [1,2], we use the word "agent" to describe a model that autonomously reasons and improves through reinforcement learning, even without direct physical or simulated environment interaction. Our use of "agent" highlights the model's ability to employ causal concepts, interpret causal structures, and refine its reasoning through self‑improvement signals within a causal reasoning framework.
>
> [1] Wang Q, Gao Z, Xu R. Graph agent: Explicit reasoning agent for graphs[J]. arXiv preprint arXiv:2310.16421, 2023.
>
> [2] Poesia G, Broman D, Haber N, et al. Learning formal mathematics from intrinsic motivation[J]. Advances in Neural Information Processing Systems, 2024, 37: 43032-43057.
>
> >Weakness 2: Evaluation is confined to the same synthetic distribution used for training; no results on external causal benchmarks (CLadder, CLEAR, CaLM) are reported.
>
> A2: The GRPO-trained model transfers effectively to unseen causal reasoning benchmarks, demonstrating strong generalization beyond the training tasks. As shown in Table R1, we evaluate the model on three unseen causal benchmarks — CaLM, CLadder, and CLEAR — to directly test its out-of-distribution performance. The model shows consistent improvements across all benchmarks after GRPO post-training.
>
> **Table R1**:  Model Performance on CaLM, CLadder and  CLEAR benchmark
>
> | **Acc.**        | **GRPO** | **Base** | **Best Performance Reported in the Original Paper** |
> | ------------------- | -------- | -------- | --------------------------------------------------- |
> | CaLM (Mathematical) | 0.935    | 0.272    | 0.270 (GPT-3.5-Turbo)                            |
> | CLadder             | 0.719    | 0.577    | 0.704 (GPT-4+CAUSALCOT)                            |
> | CLEAR               | 0.472    | 0.433    | 0.605 (GPT-4)                                        |
>
> > Weakness 3: No ablation on reward shaping or robustness to real data.
>
> A3: We conduct additional experiments on reward shaping and robustness to real-world data, and the results are presented in Table R2 and Table R3.
>
> For reward shaping, our RL training incorporates three types of rewards — the reasoning format reward, the JSON output format reward, and the answer reward. To examine the contribution of the reward design, we train an additional GRPO model without the reasoning format reward. The results show a slight performance degradation after removing this reward, suggesting that the strong overall performance mainly arises from the GRPO training framework itself rather than the specific reward design.
>
> For robustness to real-world data, we further evaluate the GRPO-trained LLM on the CausalProbe-E [3] benchmark, which assesses the human-like causal reasoning capability of LLMs. The results indicate that the model’s ability to understand real-world causality and draw coherent inferences remains well preserved after post-training, confirming the robustness and real-world applicability of our approach.
>
> [3] Chi H, Li H, Yang W, et al. Unveiling causal reasoning in large language models: Reality or mirage?[J]. Advances in Neural Information Processing Systems, 2024, 37: 96640-96670.
>
> **Table R2**:  Reward-shaped model's performance on CaLM and DeepCausa test set
>
> | **Acc.**      | **CaLM** | **Deconfounding** | **Insufficient** | **Omitted** | **Redundant** | **Rephrased** |
> | ------------- | -------- | ----------------- | ---------------- | ----------- | ------------- | ------------- |
> | GRPO_no_think | 0.941    | 0.834             | 0.891            | 0.651       | 0.905         | 0.868         |
>
> **Table R3**:  Models' Performance on CausalProbe-E benchmark
>
> | **Acc.**      | **GRPO**  | **Base** | **Best Performance Reported in the Original Paper** |
> | ------------- | --------- | -------- | --------------------------------------------------- |
> | CausalProbe-E | **0.806** | 0.803    | 0.750 (Claude 3.5 Opus)

---

> > ### Author Response · Authors · 2025-12-02
> > **Response (2/2)**
> >
> > > Question 1: How well would the trained model transfer to unseen causal benchmarks such as CaLM or CLadder?
> >
> > A4: Please refer to A2.
> >
> > > Question 2: Could the benchmark be extended to allow environment-level interaction (e.g., tool-based causal discovery)?
> >
> > A5: Yes, our benchmark can be naturally extended to include environment-level interactions. Our current design focuses on evaluating the causal inference capability of LLMs given explicit causal structures. To support environment-level interaction, the benchmark could be modified to provide structured data or raw observations instead of preset causal graphs, allowing LLMs to first conduct tool-based causal discovery (e.g., using external Python or symbolic engines).
> >
> > > Question 3: Does the model’s performance degrade when SCM variables have realistic semantics rather than random symbols?
> >
> > A6: Our benchmark already includes SCMs with both realistic semantics and random symbols, and the latter play an important role in improving the model’s generalization and robustness.
> >
> > Our dataset already contains two types of SCM variables—those with realistic semantics (e.g.,Blood pressure, physical health) and those represented by random symbols—allowing us to disentangle the causal reasoning ability of LLMs from potential linguistic priors.
> >
> > To explicitly assess their influence, we train the GRPO model on a subset containing only realistic-variable SCMs, and the performance decreased across multiple test sets compared to the mixed training setup reported in the paper. As shown in Table R4, the result indicates that random symbolic variables introduce desirable variability, reducing reliance on superficial semantics and thereby promoting model's robustness, internalization and generalization.
> >
> > **Table R4**:  Model performance of GRPO trained only with realistic-variable SCMs on CaLM and DeepCausa test set
> >
> > | **Acc.**       | **CaLM** | **Deconfounding** | **Insufficient** | **Omitted** | **Redundant** | **Rephrased** |
> > | -------------- | -------- | ----------------- | ---------------- | ----------- | ------------- | ------------- |
> > | Realistic_GRPO | 0.891    | 0.763             | 0.887            | 0.527       | 0.879         | 0.776         |
> >
> > As a final note, although our benchmark has been renamed CausaGym, we refer to it as DeepCausa in the rebuttal to maintain consistency with the reviewers’ comments.

---

### Official Review · Reviewer_dTGZ · 2025-11-04

**Soundness:** 3
**Presentation:** 2
**Contribution:** 2
**Rating:** 4
**Confidence:** 3

**Summary:**

This paper investigates whether post-training methods can enhance the causal inference capabilities of large language models (LLMs). The authors introduce DeepCausa, a comprehensive benchmark comprising seven causal tasks for training and five test sets for evaluation. They systematically compare five post-training methods—SFT, DPO, KTO, PPO, and GRPO—and demonstrate that online RL methods, particularly GRPO, significantly improve the causal reasoning abilities of smaller LLMs, enabling them to outperform larger baseline models. The study also evaluates generalization, internalization, and robustness under distribution shift and noisy data conditions.

**Strengths:**

1. The experimental design is thorough, covering multiple causal tasks, training methods, and evaluation dimensions.

2. The introduction of the DeepCausa benchmark is a valuable contribution, providing a structured dataset for training and evaluating causal inference agents.

3. The analysis is comprehensive, with clear comparisons across methods and detailed ablation studies.

**Weaknesses:**

1. The abstract could be more concise and formal. For instance, the phrase “To this end, this paper investigates whether post-training can turn LLMs into effective causal inference agents” could be rephrased to better align with academic tone.

2. The introduction repeatedly uses “we” and could be structured more objectively. For example, “We then systematically evaluate…” could be replaced with a more formal passive or impersonal construction.

3. The paper lacks a discussion on the practical utility of using LLMs for formal causal reasoning, especially given the existence of specialized causal inference tools (e.g., DoWhy, CausalML). The authors should clarify the real-world scenarios where an LLM-based agent would be preferable.

4. While the DeepCausa dataset is introduced, its naming and branding could be more distinctive (e.g., “CausalAgent-Bench” or similar) to enhance recognition and reuse.

5. There is no direct comparison with existing causal reasoning benchmarks (e.g., CLADDER, CLEAR). A dedicated section explaining how DeepCausa differs and why it is better suited for evaluating post-training methods would strengthen the contribution.

6. The figures (e.g., Figure 1) use light colors and small text, making them difficult to read. The samples are overly detailed; a more abstract and summarized visualization would improve clarity and impact.

**Questions:**

1. How does DeepCausa compare to existing causal reasoning benchmarks in terms of task coverage and difficulty?

2. Can the authors provide more insight into why online RL methods (especially GRPO) outperform offline methods so significantly?

3. What are the limitations of using synthetic SCMs for training, and how might this affect real-world applicability?

---

> ### Author Response · Authors · 2025-12-02
> **Response (1/2)**
>
> > Weakness 1: The abstract could be more concise and formal. For instance, the phrase “To this end, this paper investigates whether post-training can turn LLMs into effective causal inference agents” could be rephrased to better align with academic tone.
> >Weakness 2:  The introduction repeatedly uses “we” and could be structured more objectively. For example, “We then systematically evaluate…” could be replaced with a more formal passive or impersonal construction.
>
> A1: Thank you for your suggestion. We have updated our abstract and introduction.
>
> > Weakness 3: The paper lacks a discussion on the practical utility of using LLMs for formal causal reasoning, especially given the existence of specialized causal inference tools (e.g., DoWhy, CausalML). The authors should clarify the real-world scenarios where an LLM-based agent would be preferable.
>
> A2: Our agent is most useful when causal questions are posed by non-experts in unstructured natural language, where tabular-data tools like DoWhy or CausalML are not directly applicable. In many real-world settings, users only have unstructured textual descriptions of causal questions, before any clean tabular dataset is available for DoWhy/CausalML-style tools. Our agent enables formal causal reasoning directly from natural-language inputs without programming expertise. Our experiments also demonstrate robustness under missing or incomplete data, reflecting these practical conditions.
>
> > Weakness 4: While the DeepCausa dataset is introduced, its naming and branding could be more distinctive (e.g., “CausalAgent-Bench” or similar) to enhance recognition and reuse.
>
> A3: Thank you for your suggestion about our benchmark name, and we have changed our name to CausaGym.
>
> > Weakness 5: There is no direct comparison with existing causal reasoning benchmarks (e.g., CLADDER, CLEAR). A dedicated section explaining how DeepCausa differs and why it is better suited for evaluating post-training methods would strengthen the contribution.
>
> A4: DeepCausa differs from existing causal reasoning benchmarks by providing both a training dataset and comprehensive evaluation settings that enable systematic testing of post‑training methods in terms of generalization, internalization, and robustness. The specific comparison is shown in Table R1.
>
> Existing benchmarks such as CLADDER, CALM, and CLEAR focus mainly on evaluation and do not include corresponding training datasets. Consequently, they cannot be used to fairly compare different post‑training methods, since they do not provide a dedicated training dataset under the same framework. DeepCausa focuses on numerical causal reasoning tasks to examine how LLMs understand and apply causal concepts through computation and reasoning. By providing a dedicated training dataset, DeepCausa supports controlled post‑training experiments and allows comprehensive evaluation across three key aspects: generalization, internalization, and robustness. This integrated design makes DeepCausa a more suitable benchmark for assessing how post-training methods enhance causal reasoning.
>
> **Table R1**: Comparison between DeepCausa and existing causal reasoning benchmarks:
>
> | **Benchmark** | **Training Dataset** | **Numerical Input** | **Rationale** | **Generalization Test** | **Internalization Test** | **Robustness Test** |
> | ------------- | -------------------- | ------------------- | ------------- | ----------------------- | ------------------------ | ---------------------- |
> | CLadder       | No                   | Yes                 | Yes           | No                      | No                       | No                     |
> | CALM          | No                   | Yes                 | Yes           | No                      | No                       | No                     |
> | CLEAR         | No                   | No                  | No            | No                      | No                       | No                     |
> | DeepCausa     | Yes                  | Yes                 | Yes           | Yes                     | Yes                      | Yes                    |

---

> > ### Author Response · Authors · 2025-12-02
> > **Response (2/2)**
> >
> > > Weakness 6: The figures (e.g., Figure 1) use light colors and small text, making them difficult to read. The samples are overly detailed; a more abstract and summarized visualization would improve clarity and impact.
> >
> > A5: Thank you for your suggestion about our figures and samples, and we have updated them.
> >
> > > Question 1: How does DeepCausa compare to existing causal reasoning benchmarks in terms of task coverage and difficulty?
> >
> > A6: Please refer to A4.
> >
> > > Question 2: Can the authors provide more insight into why online RL methods (especially GRPO) outperform offline methods so significantly?
> >
> > A7: Online reinforcement learning methods, such as GRPO, outperform offline approaches because they actively collect feedback during training, allowing the model to refine and verify its reasoning process, while offline methods passively learn from limited and potentially suboptimal demonstrations.
> >
> > Unlike offline methods (such as SFT or DPO) which rely solely on static data for imitation, online RL continuously refines reasoning by using real-time, iterative reward signals to explore and correct errors. Prior studies [1] show that such dynamic feedback in online RL implicitly promotes logical consistency and correctness in multi‑step reasoning. Additionally, in the causal inference domain, high‑quality chain‑of‑thought data are scarce, so offline methods lack sufficient supervision. Therefore, the performance gap arises from both the exploratory nature of online RL, which encourages reasoning verification, and the data limitations that constrain offline approaches.
> >
> > [1] Wen X, Liu Z, Zheng S, et al. Reinforcement learning with verifiable rewards implicitly incentivizes correct reasoning in base llms[J]. arXiv preprint arXiv:2506.14245, 2025.
> >
> > > Question 3:  What are the limitations of using synthetic SCMs for training, and how might this affect real-world applicability?
> >
> > A8: Using synthetic SCMs for training does not substantially affect the model’s real-world applicability. This is supported by the model’s performance on our testing set that reflect diverse real-world scenarios.  Specifically, DeepCausa-redundant and DeepCausa-insufficient correspond to cases where users provide excessive or insufficient input, respectively, while DeepCausa-rephrasing and DeepCausa-omitted examine how the model handles variation in user expressions.
> >
> > In our experiments, the GRPO-trained model on data generated from synthetic SCMs performs consistently well across all four datasets, demonstrating strong generalization and robustness. Furthermore, we evaluate the GRPO-trained model on the CausalProbe-E [1] benchmark, which assesses LLMs’ capability for human-like causal reasoning. As shown in Table R2, the result confirms that the model’s understanding of real-world causality remains intact, further suggesting that the use of synthetic data does not compromise real-world applicability.
> >
> > [1] Chi H, Li H, Yang W, et al. Unveiling causal reasoning in large language models: Reality or mirage?[J]. Advances in Neural Information Processing Systems, 2024, 37: 96640-96670.
> >
> > **Table R2**: Model Performance on CausalProbe-E benchmark
> >
> > | **Acc.**      | **GRPO**  | **Base** | **Best Performance Reported in the Original Paper** |
> > | ------------- | --------- | -------- | --------------------------------------------------- |
> > | CausalProbe-E | **0.806** | 0.803    | 0.750 (Claude 3.5 Opus)                          |
> >
> > As a final note, although our benchmark has been renamed CausaGym, we refer to it as DeepCausa in the rebuttal to maintain consistency with the reviewers’ comments.

---

### Author Response · Authors · 2025-12-02
**General response (1/2)**

We have made a revision to our paper according to the reviewer's constructive suggestions. Below we summarize the key modifications in this revision:
- **Clarified technical contribution**：We have provided a more detailed description of our work’s contributions  (Section 1; #dTGZ) and related work (Section 4; #cKjQ).
- **Dataset detail**: We have provided examples for each test set in DeepCausa (Appendix D; #srnV), a comparison between our benchmark, the reasoning process of the post-trained LLM (Appendix I; #cKjQ) and several well-known causal reasoning benchmarks (Appendix F; #dTGZ, #3eej, #CkjQ) and training dataset statistics (Appendix G; #srnV).
- **Expanded base model**: We retrain the SFT, online RL, and offline RL models on two additional base models, and provide their evaluation results on DeepCausa (Appendix E.1; #cKjQ).
- **Added more qualitative result**: We further test performance of our GRPO model on other causal reasoning benchmarks (Appendix H; #dTGZ, #srnV) and performance of GRPO variants on DeepCausa (Appendix E.2; #3eej).

---

> ### Author Response · Authors · 2025-12-02
> **General response (2/2)**
>
> We are very encouraged by these positive comments:
> -  **Important and Timely Topic**: "The studied problem is important and practical. Both post-training methods and causal reasoning are essential for LLMs." (#cKjQ); "..., representing a non-trivial effort that provides valuable insights and lays the groundwork for further research." (#srnV)
> -  **Valuable Benchmark and Dataset Contribution**: "The introduction of the DeepCausa benchmark is a valuable contribution" (#dTGZ) "Introduces the first causal dataset that supports RL-based training with automatic rewards." (#3eej)
> - **Comprehensive and Systematic Experimental Design**: "The experimental design is thorough, covering multiple causal tasks, training methods, and evaluation dimensions." (#dTGZ); "The analysis is comprehensive, with clear comparisons across methods and detailed ablation studies." (#dTGZ); "The results analyses are comprehensive, and the authors analyze diverse aspects of agents" (#cKjQ); "The experiments are comprehensive and well-documented, offering detailed analyses across multiple aspects" (#srnV)

---

### Note · Authors · 2026-01-06

I have read and agree with the venue's withdrawal policy on behalf of myself and my co-authors.